# Prediction of Thermally Modified Wood Color Change after Artificial Weathering Based on IPSO-SVM Model

**Juncheng Li, Ning Li, Jinze Li, Wei Wang * and Haolin Wang**

College of Mechanical and Electrical Engineering, Northeast Forestry University, Harbin 150040, China; lijuncheng165134@nefu.edu.cn (J.L.)
* Correspondence: vickywong@nefu.edu.cn; Tel.: +86-133-1361-3588

**Abstract:** The support vector machine (SVM) model was applied to predict the color change of heat-modified wood after artificial weathering. In order to improve the prediction performance, the improved particle swarm optimization (IPSO) algorithm was used to optimize the parameters of the SVM model, and an improved particle swarm optimized support vector machine (IPSO-SVM) model was established on the basis of the nonlinear descending weight strategy to improve the particle swarm optimization. To verify the performance of the established model, the MAE, RMSE, and $R^2$ of the test set and training set were compared with the PSO-SVM model and the SVM model. Analysis of the results showed that compared to the PSO-SVM model and the SVM model, the IPSO-SVM model reduced the RMSE of the training set data by 49% and 72%, the MAE by 52% and 78%, the STD by 14% and 68%, the test set data by 6% and 24%, the MAE by 2% and 25%, and the STD by 22% and 29%, respectively. The results show that modeling studies using the IPSO-SVM model provide results showing that color changes in heat-modified wood after artificial weathering can be successfully predicted without expensive and time-consuming experimental studies.

**Keywords:** prediction model; particle swarm optimization; support vector machine; heat treatment of wood

## 1. Introduction

Wood has a wide variety of uses. Since ancient times, wood has been widely used in various industries such as architectural decoration, wood furniture manufacturing, and cultural and educational office supplies, and the demand for wood is growing [1]. However, it is well known that in the natural environment, due to such environmental factors as sunlight, oxygen, water, and temperature [2], the surface of the wood will change color, especially under the weathering, and the color change will be apparent to the eye [3]. Due to the photodegradation of lignin and wood extracts, the color of wood changes [4]. Moreover, after heat treatment of wood, its color will change greatly, and the demand for wood after heat treatment is increasing in various fields. The color of the wood after heat treatment is generally brown, which is similar to the color of some precious wood, and the color of the heat-treated material is uniform inside and outside, making it aesthetically pleasing [5]. In addition, compared to wood without heat treatment, heat-treated wood exposed to ultraviolet light has better color stability, which may be related to the phenol content in the wood [6]. Tomak et al. [7] believed that heat treatment could improve the color stability and surface quality of weathering samples. Studies have shown that heat-treated wood of different colors can be obtained by adjusting the temperature and time of heat treatment [8,9], and the gloss value of the wood after heat treatment is low. Due to the heat treatment, the uniformity of the color and gloss of the exposed surface of the wood improves [10], so as to realize the diversification of product varieties. Most studies have shown that the color change in heat-treated wood is highly dependent on the wood species [11]. Sikora et al. [12] studied the effect of thermal modification temperature on the color change of spruce and oak and found that the surface brightness of the wood (L*)



decreased with the increase in the treatment temperature and maximum color difference (ΔE*), and the total color difference reached its maximum value at 210 °C. The change in wood color is due mainly to the change in chemical composition. The decrease in hemicellulose content will lead to a decrease in the brightness of the wood and a change in the total color difference [13]. The reduction in the wood's brightness is also influenced by the lignin content [14].

In general, to obtain data on wood color change under different heat treatment processes it is necessary to conduct a large number of difficult and time-consuming comprehensive experiments. To obtain data more quickly and efficiently, it is necessary to explore appropriate modeling methods for the study of color change in wood after heat treatment. The principle of SVM is to minimize structural risks, so SVM is particularly effective in small-sample and nonlinear problems. Liping Sun et al. [15] proposed the least squares support vector machine (LSSVM) method to establish an online model of a wood drying system. Based on data for the deceleration drying stage obtained through a drying experiment as samples, the online prediction model of the wood drying system was established and forecasted according to the requirement of actual predictive control. Jicheng Li et al. [16] proposed a modeling method using an improved ant colony algorithm (MACA) to optimize the least squares support vector machine (LSSVM). Accurately and reliably assessing changes in wood moisture content is the key to improving wood drying quality.

In this paper, a support vector machine model is established to predict change in wood color under different artificial weathering times. The prediction model can effectively reduce the experimental cost. In this study, the prediction model is used to learn and predict the data to verify the accuracy of the model. The measurement is not carried out through real experiments, which is a limitation of this study. Therefore, the proposed model is compared with the original model and other models to verify its advantages and practicability.

## 2. Experimental Part

### 2.1. Change in Wood Color

Table 1 shows the experimental data published by Nguyen et al. [17], which can be queried on the European Journal of Wood and Wood Products website, specifically at [17] https://doi.org/10.1007/s00107-019-01449-0 (accessed 29 October 2022).

**Table 1.** Experimental measurements of color changes in heat-treated wood during artificial weathering.

| Heat-Treated Wood | Artificial Weathering Time (h) | Average of the Sample. | | | | | | | |
| | | Heat Treated Larch | | | | Heat Treated Polar | | | |
| | | ΔL | Δa | Δb | ΔE | ΔL | Δa | Δb | ΔE |
| 180 °C | 240 | −1.01 | 0.55 | −0.03 | 1.15 | 8.15 | −2.97 | −0.94 | 8.72 |
| | 480 | −0.86 | 0.25 | −0.28 | 0.94 | 6.52 | −2.13 | −0.53 | 6.88 |
| | 720 | −0.56 | −0.19 | −0.54 | 0.79 | 8.02 | −2.23 | −0.27 | 8.33 |
| | 960 | −3.22 | 0.70 | −0.41 | 3.32 | 6.19 | −0.92 | 2.33 | 6.68 |
| | 1200 | −3.88 | 1.08 | −0.4 | 4.05 | 5.92 | −0.52 | 2.05 | 6.29 |
| | 1440 | −3.19 | 1.20 | −0.53 | 3.44 | 4.15 | 0.19 | 2.89 | 5.06 |
| | 1680 | −3.58 | 1.22 | −0.38 | 4.06 | 4.34 | −0.07 | 2.88 | 5.20 |
| | 1920 | −5.09 | 1.88 | 0.41 | 5.44 | 5.48 | −0.6 | 2.86 | 6.21 |
| | 2160 | −5.19 | 2.51 | −0.25 | 5.77 | 5.57 | −0.74 | 2.46 | 6.13 |
| | 2400 | −5.00 | 2.47 | −0.45 | 5.59 | 5.90 | −1.14 | 0.98 | 6.09 |
| | 2700 | −4.68 | 1.73 | −0.57 | 5.02 | 5.47 | −2.55 | −0.48 | 6.05 |
| | 3000 | −3.76 | 1.45 | −1.28 | 4.23 | 5.24 | −2.75 | −0.78 | 5.97 |
| 190 °C | 240 | −2.73 | 1.02 | 0.71 | 3.00 | 7.73 | −2.21 | 0.15 | 8.04 |
| | 480 | −2.12 | 0.73 | 0.80 | 2.38 | 14.30 | −1.69 | 2.10 | 6.58 |
| | 720 | −2.03 | 0.38 | 0.80 | 2.22 | 14.63 | −1.37 | 3.05 | 7.89 |
| | 960 | −3.33 | 1.04 | 0.83 | 3.58 | 5.00 | 0.01 | 6.06 | 7.85 |

**Table 1.** *Cont.*

| Heat-Treated Wood | Artificial Weathering Time (h) | Average of the Sample. | | | | | | | |
|---|---|---|---|---|---|---|---|---|---|
| | | Heat Treated Larch | | | | Heat Treated Polar | | | |
| | | ΔL | Δa | Δb | ΔE | ΔL | Δa | Δb | ΔE |
| | 1200 | −3.81 | 1.21 | 0.71 | 4.05 | 4.70 | 0.57 | 6.34 | 7.91 |
| | 1440 | −3.24 | 1.26 | 0.85 | 3.57 | 5.72 | 0.57 | 6.93 | 9.01 |
| | 1680 | −3.63 | 1.74 | 0.87 | 4.12 | 5.77 | 0.58 | 7.67 | 9.62 |
| | 1920 | −4.28 | 1.78 | 1.27 | 4.81 | 5.81 | 0.64 | 9.17 | 10.88 |
| | 2160 | −4.44 | 2.03 | 1.43 | 5.09 | 6.05 | 1.61 | 9.39 | 11.28 |
| | 2400 | −3.94 | 1.90 | 1.37 | 4.58 | 6.57 | 1.27 | 8.60 | 10.90 |
| | 2700 | −3.66 | 1.77 | 1.05 | 4.20 | 6.86 | 0.61 | 8.09 | 10.62 |
| | 3000 | −3.09 | 1.35 | 0.82 | 3.47 | 6.33 | 0.17 | 6.96 | 9.41 |
| 200 °C | 240 | 0.54 | 1.12 | 0.17 | 1.26 | 13.40 | −3.67 | 0.10 | 8.36 |
| | 480 | 1.00 | 1.05 | 0.35 | 1.49 | 5.09 | −0.07 | 4.10 | 8.53 |
| | 720 | 1.95 | 0.38 | 0.31 | 2.01 | 7.52 | −1.07 | 3.69 | 8.44 |
| | 960 | 0.30 | 1.09 | 0.79 | 1.38 | 8.60 | −0.92 | 5.06 | 10.02 |
| | 1200 | 0.50 | 1.46 | 1.53 | 2.17 | 6.45 | 0.27 | 5.57 | 8.52 |
| | 1440 | 0.14 | 1.29 | 1.02 | 1.65 | 7.50 | 0.31 | 5.81 | 9.49 |
| | 1680 | 0.07 | 2.02 | 1.87 | 2.75 | 8.44 | −0.02 | 6.24 | 10.50 |
| | 1920 | −0.04 | 2.17 | 2.96 | 3.67 | 9.79 | −0.35 | 6.78 | 11.91 |
| | 2160 | 0.11 | 2.22 | 3.63 | 4.26 | 9.90 | −0.42 | 6.82 | 12.03 |
| | 2400 | 0.27 | 2.20 | 3.31 | 3.98 | 10.12 | −0.65 | 6.03 | 11.80 |
| | 2700 | 0.29 | 1.75 | 2.62 | 3.17 | 10.35 | −0.86 | 5.34 | 11.68 |
| | 3000 | 0.85 | 1.28 | 1.52 | 2.16 | 9.91 | −0.97 | 4.68 | 11.00 |
| 210 °C | 240 | 0.72 | 2.04 | 6.07 | 6.44 | 7.74 | −0.62 | 2.19 | 8.07 |
| | 480 | 4.04 | 2.04 | 7.89 | 9.10 | 3.85 | 1.33 | 3.17 | 5.16 |
| | 720 | 2.98 | 2.69 | 8.66 | 9.54 | 5.44 | 0.95 | 3.79 | 6.70 |
| | 960 | −1.09 | 3.16 | 7.04 | 7.80 | 3.48 | 2.19 | 6.00 | 7.28 |
| | 1200 | −0.05 | 3.23 | 6.74 | 7.48 | 2.92 | 2.83 | 6.52 | 7.68 |
| | 1440 | 0.55 | 3.16 | 7.20 | 7.88 | 4.77 | 2.88 | 7.17 | 9.08 |
| | 1680 | 0.27 | 3.47 | 7.20 | 8.00 | 4.08 | 2.92 | 7.65 | 9.15 |
| | 1920 | −0.59 | 3.73 | 7.62 | 8.51 | 3.96 | 3.39 | 8.64 | 10.09 |
| | 2160 | −0.57 | 3.77 | 7.90 | 8.77 | 3.18 | 3.48 | 9.09 | 12.23 |
| | 2400 | −0.35 | 3.66 | 7.77 | 8.60 | 3.69 | 3.01 | 8.41 | 9.66 |
| | 2700 | −0.25 | 3.56 | 7.65 | 8.44 | 4.00 | 2.86 | 8.22 | 9.58 |
| | 3000 | 0.27 | 3.23 | 7.07 | 7.78 | 3.36 | 2.83 | 7.97 | 9.10 |
| 220 °C | 240 | −3.42 | 2.19 | 4.31 | 5.92 | 6.13 | −0.39 | 3.33 | 6.99 |
| | 480 | −2.52 | 3.64 | 5.38 | 6.50 | −0.22 | 2.04 | 3.87 | 4.38 |
| | 720 | −1.43 | 2.93 | 5.85 | 6.69 | 2.65 | 1.82 | 4.92 | 5.88 |
| | 960 | −3.45 | 2.73 | 5.00 | 6.66 | −0.13 | 2.71 | 5.82 | 6.42 |
| | 1200 | −2.37 | 2.59 | 4.55 | 5.75 | 1.19 | 2.74 | 5.54 | 6.29 |
| | 1440 | −2.26 | 2.64 | 4.32 | 5.55 | 2.47 | 2.86 | 5.77 | 6.90 |
| | 1680 | −2.17 | 2.90 | 4.70 | 5.94 | 2.43 | 2.92 | 5.84 | 6.96 |
| | 1920 | −2.02 | 3.39 | 5.30 | 6.60 | 1.99 | 3.38 | 7.87 | 8.79 |
| | 2160 | −1.71 | 3.53 | 5.64 | 6.87 | 1.71 | 3.66 | 11.38 | 12.08 |
| | 2400 | −1.64 | 3.41 | 5.41 | 6.60 | 1.93 | 3.54 | 10.94 | 11.66 |
| | 2700 | −1.47 | 3.23 | 5.30 | 6.38 | 3.07 | 3.38 | 10.50 | 11.45 |
| | 3000 | −0.93 | 3.00 | 4.96 | 5.87 | 2.71 | 3.22 | 9.98 | 10.83 |

In this experiment, two different types of wood, cork and hardwood, were selected for heat treatment processing; the specific species were heat-treated larches (*Larix gmelini*) and heat-treated poplar (*Populus alba*), and they were provided by the College of Materials Science and Engineering of Northeast Forestry University. The heat treatment time for both wood types was four hours, and the heat treatment temperature was 180, 190, 200, 210, and 220 °C, respectively. One-hundred-five pieces of each of the two types of wood were randomly divided into ten groups of 21 samples each. The sample size was $80 \times 30 \times 3$ mm$^3$ (l × t × r). The sample was placed at a room temperature of $20 \pm 2$ °C and a relative

humidity of $65 \pm 5\%$ until the moisture content of the sample was about 12%. The samples were subjected to an accelerated weathering test in a weathering tester to simulate the damage to the wood that would normally take months or years. The weathering time was 0, 240, 480, 720, 960, 1200, 1440, 1680, 1920, 2160, 2400, 2700, and 3000 h, respectively. Color variations on the sample area were evaluated. The color change was measured using the CIE 1976 L\*a\*b\* color measurement system [18], and the total color change ($\Delta E$) was calculated as follows:

$$\Delta L^* = L_2{}^* - L_1{}^* \tag{1}$$

$$\Delta a^* = a_2{}^* - a_1{}^* \tag{2}$$

$$\Delta b^* = b_2{}^* - b_1{}^* \tag{3}$$

$$\Delta E = \sqrt{\Delta L^{*2} + \Delta a^{*2} + \Delta b^{*2}} \tag{4}$$

where $\Delta L^*$, $\Delta a^*$, and $\Delta b^*$ represent color coordinate changes; $L_1{}^*$, $a_1{}^*$, $b_1{}^*$ indicates the brightness, red and green coordinates, and yellow and blue coordinates of the untreated sample; and $L_2{}^*$, $a_2{}^*$, $b_2{}^*$ represents the brightness, red and green coordinates, and yellow and blue coordinates of the treated sample.

As Figure 1a shows, the untreated larch $\Delta E$ increased before 1680 h and decreased from 1680 h to 3000 h. According to Figure 1b, the $\Delta E$ of the untreated poplars showed an upward trend before 1200 h, and then a downward trend, which was consistent with the results of Xing et al. [19]. The $\Delta E$ of the two untreated woods was greater than that of the heat-treated woods, suggesting that the heat-treated artificially weathered woods had better color stability.

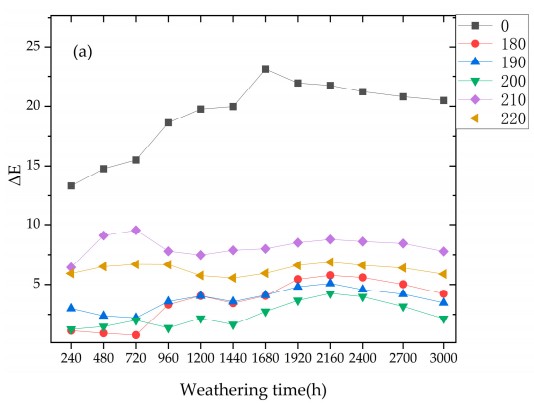 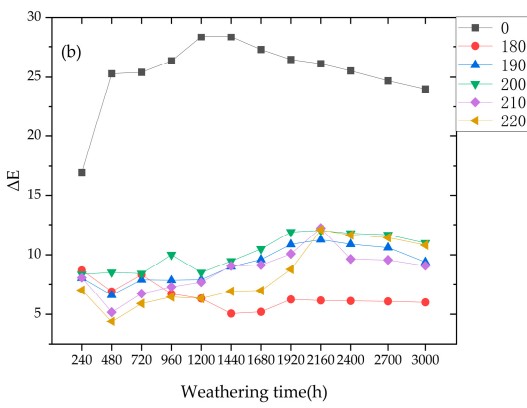

**Figure 1.** (**a**) $\Delta E$ of heat-treated larch and (**b**) $\Delta E$ of heat-treated poplar.

*2.2. Prediction Model*

2.2.1. Support Vector Machine Model

The Support Vector Machine (SVM) was initially devised by Vapnik, a former Soviet professor [20]. The SVM regression model has good adaptability in solving linear and nonlinear prediction and can better solve the complicated problems of influencing factors in the prediction of wood color change in the process of wood heat treatment. Support vector machines have performed well for small-sample, nonlinear, and large dimensional problems and are widely used to treat classification and regression problems. Finally, support vector machines can be transformed into a dual optimization form [21]. This is given in Formulas (5) and (6):

Objective function:

$$min\frac{1}{2}\sum_{ij=1}^{1}(\alpha_i^* - \alpha_i)(\alpha_j^* - \alpha_j)K(x_i, x_j) + \varepsilon\sum_{i=1}^{1}(\alpha_i^* - \alpha_i) - \sum_{i=1}^{1}y_i(\alpha_i^* - \alpha_i) \tag{5}$$

Conditions:

$$\begin{cases} \sum\limits_{i=1}^{1}y_i(\alpha_i - \alpha_i^*) = 0 \\ \alpha_i, \alpha_i^* \geq 0, i = 1, 2, \ldots l \end{cases} \tag{6}$$

In Equations (5) and (6), $\alpha_i$ and $\alpha_i^*$ are Lagrange multipliers, and $K(x_i, x_j)$ is the kernel function. Commonly used kernel functions include the linear kernel function (LinearKernel, LK), the polynomial kernel function (PolynomialKernel, PK), and the radial basis kernel function (RBF), as shown in Table 2. For nonlinear problems, the kernel function is introduced, the input space is transformed into a high-dimensional space through nonlinear mapping, and the nonlinear problem becomes a linear problem. Because RBF has strong nonlinear mapping ability, it can map the original features to infinite dimensions, which has been widely used at present. RBF is adopted as the kernel function in this paper. RBF can be expressed by Formula (7):

$$K(x_i, x_j) = \exp\left(-\delta \cdot |x_i - x_j|^2\right) \tag{7}$$

where $\delta$ is the kernel function parameter representing the space range that a specific training sample can reach.

**Table 2.** Model parameters.

| Algorithm | Parameter |
|---|---|
| SVM | c = 4.0; g = 0.8 |
| PSO-SVM | c1 = 1.5; c2 = 1.7; maxgen = 50; sizepop = 5; popcmax = 100; popcmin = 0.1 popgmax = 100; popgmin = 0.1 |
| IPSO-SVM | c1 = 1.5; c2 = 1.7; maxgen = 50; sizepop = 5; popcmax = 100; popcmin = 0.1 popgmax = 100; popgmin = 0.1; w_star = 2; w_end = 0.4 |

After the kernel function is determined, in order to optimize the predictive performance of the SVM model, two optimal parameters, namely penalty factor c and kernel parameter g, should be found. In order to determine the c and g of the SVM reasonably quickly, this paper introduces an improved PSO to optimize the c and g of SVM and presents an improved particle swarm optimization support vector machine (IPSO-SVM) algorithm.

### 2.2.2. Improved Particle Swarm Optimization

Based on simulated social behaviors such as bird feeding and human cognition, J. Kennedy et al. [22] proposed a global optimization algorithm–particle swarm optimization (PSO), which is an evolutionary computing technology based on swarm intelligence. The particle swarm optimization algorithm has the advantages of simplicity, easy implementation, few parameters to be adjusted, and no gradient information. However, since all the particles fly toward the optimal region, the diversity will be lost and the particles will tend to be homogeneous, which causes the convergence speed to be obviously slow in the late stage of convergence. At the same time, when the algorithm converges to a certain degree, it cannot continue to optimize, and it is easy to fall into local optimal, so the convergence accuracy is not high. Therefore, many scholars are committed to improving the convergence accuracy of the PSO algorithm. In view of the above limitations, Shi Y et al. [23] in 1998 proposed linear decreasing weight strategy (LDIW), an improved particle swarm optimization algorithm with inertial weights. At the same time, aiming at the problems of premature convergence and non-convergence in the particle swarm optimization algorithm, this paper

proposes an improved particle swarm optimization algorithm with nonlinear decreasing inertia weight. In the algorithm iteration process, the particle boundary velocity is limited by the maximum velocity nonlinear decreasing change strategy. Nonlinear decreasing variation in inertia weight is used to balance the global research capacity at the early stage and the local optimization at the late stage. It can be expressed by Formula (8):

$$w = w_s - (w_s - w_e)\left| f_1 \sqrt{\frac{t}{T_{max}}} \right| \tag{8}$$

where $w_s$ is the initial inertia weight; $w_e$ is the end value of the maximum number of iterations allowed; $T_{max}$ is the maximum number of evolutions; $t$ is the number of current iterations; and $f_1$ is the regulating factor, in order to control the rate of change of $w$. The root sign in the formula is to generate the square to form the nonlinear effect, and the absolute value is added to prevent the negative sign of the open square root. As the iteration progresses, the inertia weight decreases nonlinearly. The inertia weight $w$ takes a larger value in the algorithm of early iteration so as to avoid the algorithm falling into the local extreme value and to maintain a strong global search ability. At the same time, a small inertia weight is selected in the late iteration to enhance the local search ability and accelerate the convergence speed.

### 2.2.3. IPSO-SVM Model

This paper presents an improved particle swarm optimization (IPSO) algorithm to optimize the support vector machine model. As a machine learning algorithm, SVM has unique advantages in dealing with nonlinear relation. Through nonlinear mapping, complex nonlinear problems can be transformed into linear regression problems in high dimensional space, and the complex relationship between multiple factors can be quickly learned and analyzed. The essence of the support vector machine (SVM) regression prediction model is to find the optimal relationship between each input and output sample on the basis of the limited sample data and give a reasonable output for the input samples in the test set. In order to optimize the predictive performance of the SVM model, IPSO is used to optimize and improve the SVM model. The process is as follows: First, determine the input and output of the model, divide the sample data into two categories—training and testing—and normalize all the sample data. Then the improved particle swarm optimization algorithm is used to optimize the population parameters and set the population size, nonlinear decreasing weight factor, iteration times, and other parameters. Next, the particle fitness value is calculated and the particle velocity, position, and other parameters are updated to determine the end conditions of the optimal parameters. Finally, the IPSO-SVM regression prediction performance is tested through the test set data. If the predicted value meets the set requirements, the predicted value is output. Otherwise, continue optimizing the parameters until the end condition is met. This process is shown in Figure 2:

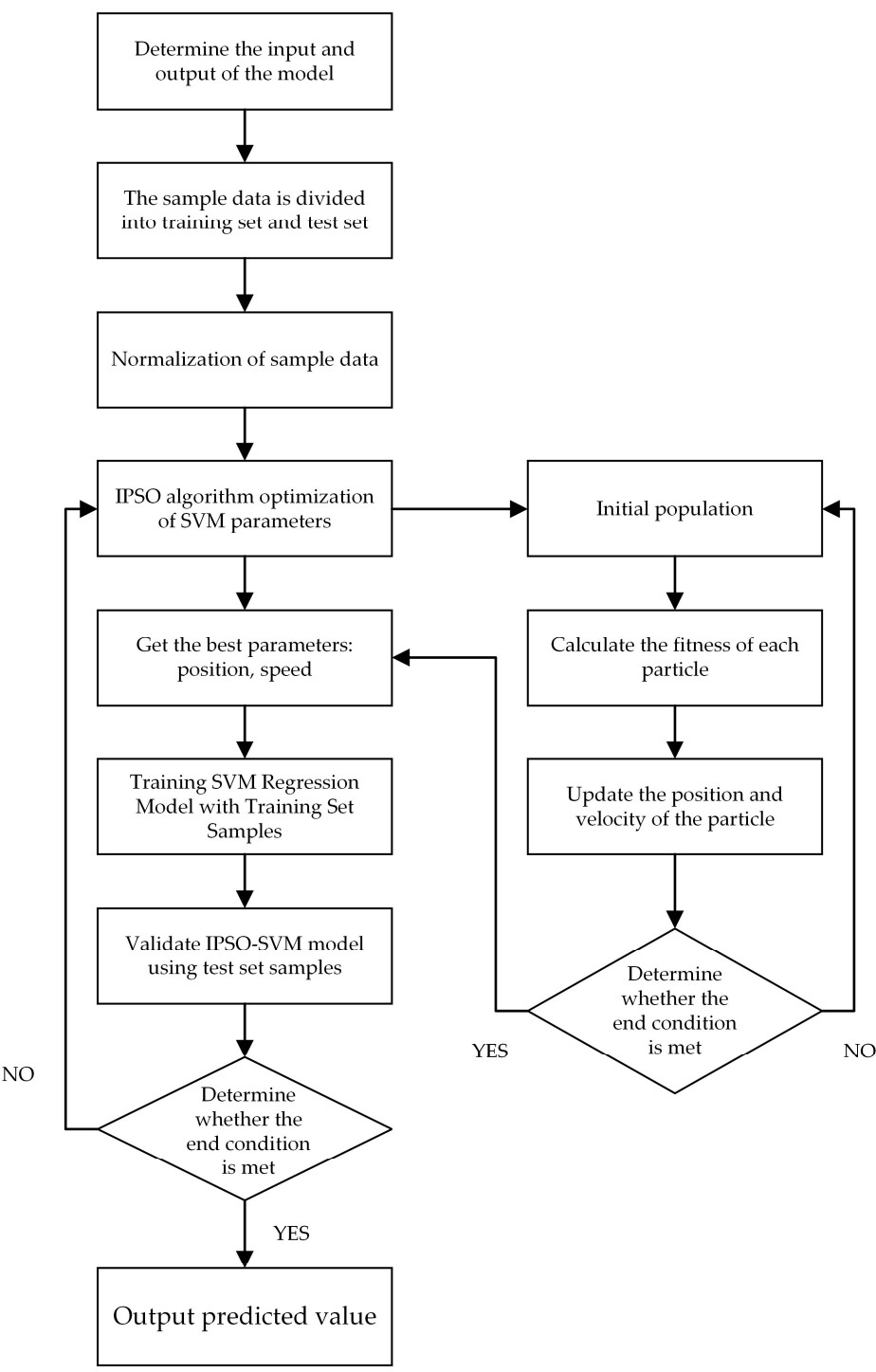

**Figure 2.** IPSO-SVM process diagram.

## 3. Results and Discussion

### 3.1. Model Parameter Settings

In order to verify the predictive performance of the proposed model, the improved particle swarm optimization support vector machine model (IPSO-SVM) was compared with the particle swarm optimization support vector machine model (PSO-SVM) and the support vector machine model (SVM). The parameters of each model are shown in Table 2. In the SVM model, c is the penalty factor and g is the radial basis function parameter. In the IPSO-SVM model, c1 represents the local search capability of the PSO parameters, c2 represents the global search capability of the PSO parameters, maxgen is the maximum

number of evolution, sizepop is the maximum number of population, popcmax is the maximum change value of SVM parameter c, popcmin is the minimum change value of SVM parameter c, popgmax is the maximum change value of SVM parameter g, popgmin is the minimum change value of SVM parameter g, and w_star and w_end are the initial and end values of the nonlinear decreasing weight factors. In the PSO-SVM model, parameter representation is the same as that in the IPSO-SVM model.

### 3.2. Model Evaluation Criteria

Common regression evaluation indexes include mean absolute error (MAE), mean square error (MSE), root mean square error (RMSE), goodness of fit ($R^2$), and standard deviation (STD), where RMSE is the square root of the MSE and the order of magnitude is the same as the true value. MSE and RMSE are essentially the same, but RMSE is used for better data description. Therefore, RMSE only is used as the evaluation index in this paper. In order to illustrate the degree of dispersion of the samples, the smaller the RMSE, the better. $R^2$ reflects the accuracy of the model fitting data. Generally, $R^2$ varies in the range of 0 to 1. The closer the value is to 1, the stronger the fitting ability is and the better the fitting effect is; otherwise, the worse the fitting effect is. Formulas (9)–(12) are shown as follows:

$$\text{RMSE} = \sqrt{\frac{\sum_{i=1}^{N}(A_i - F_i)^2}{N}} \tag{9}$$

$$\text{MAE} = \frac{1}{N}\sum_{i=1}^{N}|A_i - F_i| \tag{10}$$

$$R^2 = 1 - \frac{\sum(F_i - A_i)^2}{\sum(F_i - A_i)^2} \tag{11}$$

$$\text{STD} = \sqrt{\frac{\sum_{i=1}^{N}(x_i - \bar{x})^2}{N-1}} \tag{12}$$

where $A_i$ and $F_i$ denote the actual and predicted values, respectively, $x_i$ denotes the error between the predicted and actual values, and $\bar{x}$ denotes the average of all the errors.

### 3.3. Comparative Analysis of Model Performance

This paper compares the IPSO-SVM model with the PSO-SVM model and the SVM model. The evaluation results of each model are listed in Table 3.

**Table 3.** Model evaluation results.

| Model | | IPSO-SVM | PSO-SVM | SVM |
|---|---|---|---|---|
| RMSE | Training | 0.37363 | 0.73098 | 1.3534 |
| | Testing | 0.9166 | 0.97504 | 1.2056 |
| MAE | Training | 0.21446 | 0.44525 | 0.95681 |
| | Testing | 0.73084 | 0.7491 | 0.9691 |
| $R^2$ | Training | 0.98288 | 0.9369 | 0.77346 |
| | Testing | 0.90127 | 0.87627 | 0.83279 |
| STD | Training | 0.43294 | 0.64479 | 1.35062 |
| | Testing | 0.8536 | 1.09641 | 1.20145 |

It can be seen from Table 3 that the RMSE of the training set and the test set of the IPSO-SVM model are 0.37363 and 0.9166, the MAE values are 0.21446 and 0.73084, and the STD values are 0.43294 and 0.8536, respectively. In addition, the $R^2$ of the training set and of the test set is 0.98288 and 0.90127, respectively. It is proved that the measured results have a good fit with the model prediction. Compared to the PSO-SVM model and the SVM model, the IPSO-SVM model reduced the RMSE of the training set data by 49% and 72%, the MAE by 52% and 78%, the STD by 14% and 68%, the test set data by 6% and

24%, the MAE by 2% and 25%, and the STD by 22% and 29%, respectively. This indicates that the optimization effect of the IPSO-SVM model is obvious, and the prediction effect of IPSO-SVM model is better than that of the other two models.

Figure 3 shows the fitness convergence curves of the IPSO-SVM model and the PSO-SVM model. It can be seen from Figure 3 that the IPSO-SVM model reaches the optimum in the 27th generation while the PSO-SVM model reaches the optimum in the 34th generation. Moreover, the fitness value of the IPSO-SVM model is always smaller than that of the PSO-SVM model. It can be proved that the IPSO-SVM model is superior to the PSO-SVM model in terms of convergence speed and convergence accuracy.

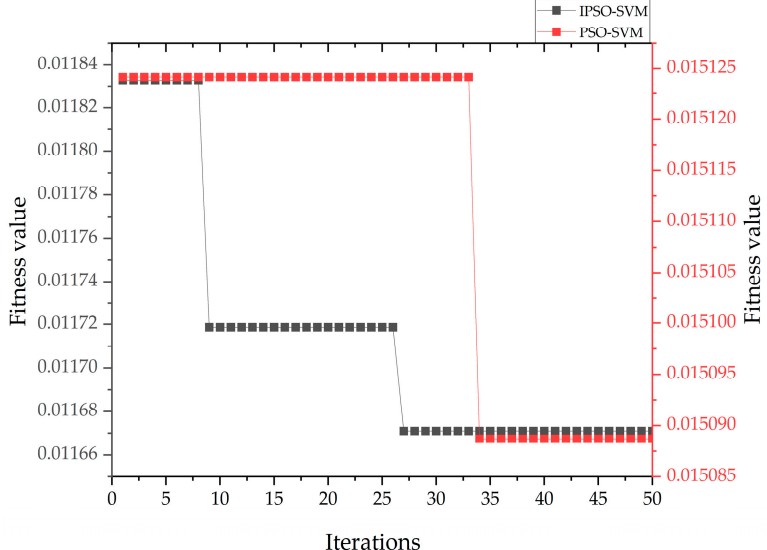

**Figure 3.** Fitness convergence curves of IPSO-SVM and PSO-SVM models.

The predicted effect of the model can also be observed in the line graph. Figure 4 shows the predicted results of the three prediction models. Compared with the other three models, the RMSE and MAE of the IPSO-SVM model are significantly lower than those of the other three models, and the $R^2$ of the IPSO-SVM model is closer to 1 than that of the other three models, indicating the highest goodness of fit. From Figure 4 combined with Table 3, it can be clearly seen that the predicted effect of the IPSO-SVM model is significantly better than that of the other two prediction models, and the predicted effect from good to bad is represented by the sequence of IPSO-SVM model, PSO-SVM model, and SVM model.

Concerning the thermal modification of artificial weathering wood color change to $R^2$ of the values listed in Table 3, we confirmed the IPSO-SVM model between the prediction results and the real value as having a very good fitting effect. The results show that the IPSO-SVM model for predicting thermal modification after artificial weathering wood color change is very accurate. a and b in Figure 4 show the comparison between the actual value and the predicted value of the color change under heat treatment by the IPSO-SVM model. Obviously, compared with other models, the actual value matches the predicted value better. Therefore, after the IPSO-SVM model is trained by the training set, the prediction effect of this model is satisfactory, and the prediction result is also within the acceptable range and can be used to predict the change in wood color after artificial weathering.

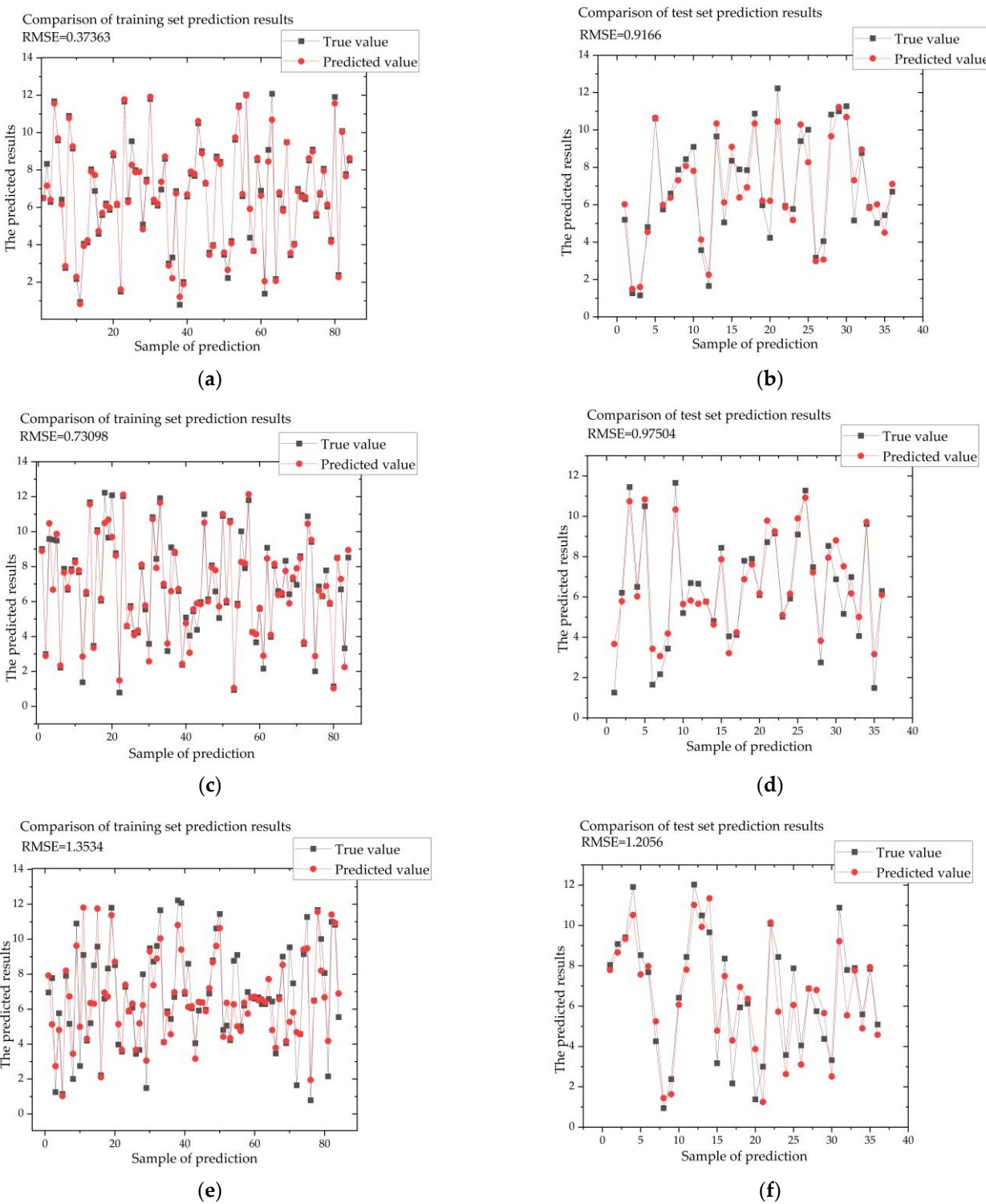

**Figure 4.** (**a**,**b**) are the prediction results of the IPSO-SVM model, (**c**,**d**) are the prediction results of the PSO-SVM model, (**e**,**f**) are the prediction results of the SVM model.

## 4. Conclusions

In order to solve the problems of wood color prediction, an SVM prediction model based on improved particle swarm optimization was proposed. In this study, the color change of heat-modified wood after artificial weathering was predicted in larch (*Larix gmelini*) and poplar (*Populus alba*). The heat treatment temperature, artificial weathering time, and wood type were used as the input variables, and the total color difference (ΔE) was used

as the output variable to predict the color change. In order to verify the performance of the IPSO-SVM model in predicting wood color change, the mean absolute error (MAE), root mean square error (RMSE), goodness of fit ($R^2$), and standard deviation (STD) of the SVM model and of the PSO-SVM model were respectively compared. It was found that the IPSO-SVM model had the highest correlation coefficient and the lowest root mean square error, mean absolute error, and standard deviation. This shows that the IPSO-SVM model has the best predictive effect. Modeling studies using the IPSO-SVM model have provided results showing that the color changes of heat-modified wood after artificial weathering can be successfully predicted.

**Author Contributions:** Conceptualization, J.L. (Juncheng Li) and W.W.; methodology, J.L. (Juncheng Li); software, N.L.; validation, J.L. (Juncheng Li), W.W. and H.W.; formal analysis, J.L. (Juncheng Li); investigation, H.W.; resources, J.L. (Juncheng Li); data curation, J.L. (Juncheng Li); writing—original draft preparation, J.L. (Juncheng Li); writing—review and editing, J.L. (Juncheng Li); visualization, J.L. (Jinze Li); supervision, H.W.; project administration, W.W.; funding acquisition, W.W. All authors have read and agreed to the published version of the manuscript.

**Funding:** This study was funded by the Fundamental Research Funds for the Central Universities, grant number 2572019BL04, and the Scientific Research Foundation for the Returned Overseas Chinese Scholars of Heilongjiang Province, grant number LC201407.

**Institutional Review Board Statement:** Not applicable.

**Informed Consent Statement:** Not applicable.

**Data Availability Statement:** The data from this paper are openly available in a public repository that issues datasets with DOIs. The data that support the findings of this study are openly available in the European Journal of Wood and Wood Products at https://doi.org/10.1007/s00107-019-01449-0, reference number [17] (accessed on 29 October 2022). The data presented in this study are available in the article.

**Conflicts of Interest:** The authors declare no conflict of interest.

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
