# Peer review of "Prediction of Thermally Modified Wood Color Change after Artificial Weathering Based on IPSO-SVM Model"

_forests, doi:10.3390/f14050948_

Round 1

Reviewer 1 Report

Comments for authors

The topic of this paper is appropriate for the journal. My suggestions are listed below:

Title :

·      Page 1 Line 3 : Define abbreviations of IPSO-SVM

Abstract :

·      Please add the conclusion in the end of abstract

Introduction :

·      Page 1 Line 29-30 : The reference was not found. Please check again.

·      The paper focus on thermal modified wood color changes. Recently, many papers are available on thermal modification. In introduction, please kindly consider this following important recent review papers:

10.4067/S0718-221X2019005000208

Change of wood color :

·      Page 4 line 94 : The reference was not found. Please check again.

·      Page 3 line 83-84 : Please use italic for Latin names, Larix gmelina and Populus alba

·      Page 3 line 87 : Please correct mm3 to mm3

·      Equation 1-4 : Please define L1, L2, a1, a2, b1, b2

·      Please define equation 5

Prediction model :

·      Page 5 line 118-119 : The reference was not found. Please check again.

·      Page 6 line 176 and 178 : Please define abbreviations at only first occurrence, you don’t need to write completely after you define in the beginning

Conclusion :

·      In practical aspect, is the IPSO SVM model only able to predict the color change after weathering?

References :

·      Page 11 line 135 : Please check how to write the authors as a journal template.

I don't have any issues for English language. Minor editing in English language

Author Response

Dear reviewer, thank you very much for your suggestions on my article. I have made modifications to your suggestions in the article. In addition, to your questions about the conclusion, my answer is that this model has been adjusted for this data, and may not have a good effect on other data. If other data are to be predicted, some adjustments may be needed. And I think it is very useful for the literature reference suggestions you put forward.

Reviewer 2 Report

Reviewer #: Manuscript Number: forests-2358894

Full Title: Prediction of Thermal Modified Wood Color Change after Arti-2 ficial Weathering Based on IPSO-SVM Model

Article Type: Original Research

Keywords: Prediction model; Particle swarm optimization; Support vector machine; Heat treatment of wood

Dear Editor,

Minor Revision

The overall presentation is average, while the authors should clarify the following issue before the paper can be accepted for publication.

The aim of the study “a support vector machine model is established to predict the change of wood color under different artificial weathering times”

Line 26. the natural environment, due to the environment (sunlight, oxygen, water, temperature) [Citation needed] The work suggested below may contribute.

A Can, H Sivrikaya Surface characterization of wood treated with boron compounds combined with water repellents, Color Research and Application 44 (3), 462-472

Line 27. microorganisms (fungi) [Citation needed]

Line 28. …. especially under the weathering, the color change will be more obvious. [citation needed] The work proposed below may contribute.

H Sivrikaya, A Can, İ Tümen, D Aydemir, Weathering Performance of Wood Treated with Copper Azole and Water Repellents, Wood research 62 (3), 437-450

Line 29. [Error! Reference 29 source not found.].?? [Citation needed]

Line 36. Believed.. Letter B should be written in lower case

Line 83-84. Populus Alba and Larix gmelini should be written in italics. Also all your net should be checked

Line 85. It should be stated how many samples were used.

Line 118. [Error! Reference source not found.] . [Citation needed]

Good

Author Response

Dear reviewer, thank you very much for your suggestions on my article. I have made modifications to your suggestions in the article. And I think it is very useful for you to give some suggestions on literature reference.

Reviewer 3 Report

The manuscript presents research on the Prediction of Thermal Modified Wood Color Change using a mathematical model. In my opinion, the article is interesting for the prediction of wood materials, but some aspects can be improved.

Line 7: “to the prediction of …” change to “to predict…”.

Lines 13-16: “According…respectively.” The subject of the sentence is not clear.

Line 25: “…is also growing.”. I advise enhancing the literature to support the sentence. Some suggestions:

https://doi.org/10.1007/s11696-021-01756-y and https://doi.org/10.1007/s13762-013-0222-3

Line 29 and 118: correct the “Error…”.

Lines: 36, 55, 60 and 63: attention on the use of uppercase after “et al.”.

Line 49: The decrease in brightness is also influenced by the lignin content. I suggest also adding this information to the manuscript. Some literature in support of the sentence: https://doi.org/10.1007/s11696-022-02181-5 and https://doi.org/10.3182/20050703-6-CZ-1902.01655

Line 73: I advise adding the “2. Experimental part” and changing headings “2.1 Change of wood color”, and “2.2 Prediction Model”.

Line 196: Figure 2, written part in the rhombus figures.

Line 234: Table 3, do you have data about the Standard Deviation?

Line 240-243: “Compared…respectively.”. As in the abstract, this sentence is unclear; please modify it.

In my opinion, the text is fluent. I have just some observations that I reported in my list of comments (Line 7, Lines 13-16 and Lines 240-243).

Author Response

Dear reviewer, thank you very much for your suggestions on my article. I have made modifications to your suggestions in the article. Among them, I think your suggestion on reference is very useful. Besides, I have added the data of standard deviation in Table 3 and modified the relevant content in the paper.
